# DiffSim: Aligning Diffusion Model and Molecular Dynamics Simulation for Accurate Blind Docking

## Abstract

Predicting the ligand's binding conformation within a target protein is a pivotal step in drug discovey. Based on prior knowledge of the binding site (protein pocket) on the target protein, biochemical researchers use molecular docking software to generate the ligand conformation within that pocket. Despite its speed, molecular docking is ill-suited for blind docking where the pocket is unknown, and the generated ligand conformation often lacks required precision. Recently, deep generative models, especially diffusion models, have been proposed for accurate blind docking. However, it is found that while deep generative models excel in locating the pocket, they still lag behind traditional methods in terms of conformation generation. Thus, bridging such gap with a hybrid approach is naturally expected to further improve the model performance. Therefore, in this study, we introduce a blind docking approach named DiffSim to seamlessly integrate the diffusion model with molecular dynamics (MD) simulation. We propose a novel loss function to align reverse diffusion sampling with MD simulation trajectories, aiming to efficiently generate ligand conformations informed by MD-modelled protein-ligand interactions at atomic resolution. Through theoretical analysis, we unveil the consistency in dynamics between diffusion models and MD simulation, demonstrating that the diffusion model is essentially a coarse-grained simulator for MD simulation. Empirical results demonstrate the effectiveness of our approach and highlight the potential of combining physics-informed MD simulation with deep learning models in drug discovery.

## 1 Introduction

Identifying bioactive ligands for a specific target within a vast compound library is a significant challenge in drug development, made laborious and costly by the need to experimentally test thousands of compounds (Blanes-Mira et al., 2022). To reduce the overall cost of drug discovery, molecular docking (Raval & Ganatra, 2022) serves as an efficient technique to computationally predict binding conformations of small molecules within the binding site (also called protein pocket) of a target protein, drastically reducing the conformation space that otherwise needs to be explored through time-consuming and cost-intensive experiments. However, scoring functions used in molecular docking often rely on simplifications about molecular interactions (Singh et al., 2022), which are inadequate for modeling intricate protein-ligand interactions and may lead to inaccurate ligand conformation predictions. In addition, traditional molecular docking methods require prior knowledge of the approximate binding site on the target protein, which makes them ill-suited for blind docking where the binding site is unknown. To overcome the limitations of traditional molecular docking, machine (deep) learning models have emerged, aiming to achieve accurate blind docking. Numerous models focus on tackling particular subtasks of blind docking, including pocket searching (Krivák & Hoksza, 2018; Jiménez et al., 2017; Yan et al., 2022), conformation generation (Xu et al., 2022; Zhu et al., 2022), and scoring functions (Wang & Zhang, 2017; Liao et al., 2019; Shen et al., 2022). Some latest models (Zhang et al., 2022; Lu et al., 2022; Stärk et al., 2022; Corso et al., 2022) have been dedicated to addressing the entire process of blind docking, including all subtasks in an end-to-end manner, which can directly generate ligand conformations conditioned on target proteins.

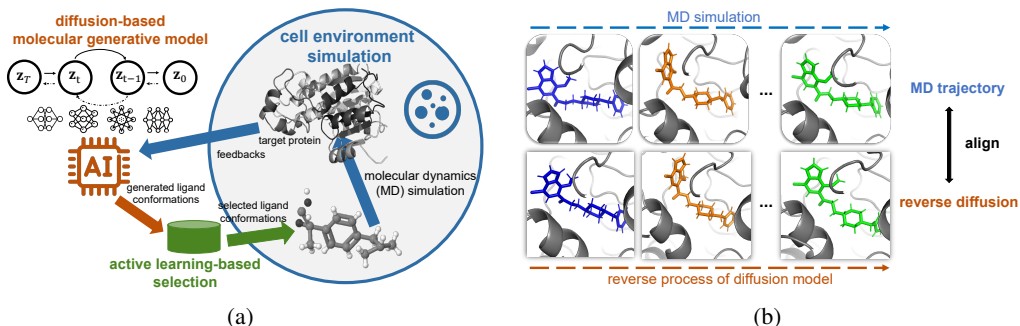

Figure 1: Overview of DiffSim. **(a)** The figure illustrates the fundamental concept behind DiffSim, a hybrid approach seamlessly integrating the diffusion model with molecular dynamics (MD) simulation. **(b)** The figure illustrates the specific approach of DiffSim: aligning reverse diffusion with MD trajectories to achieve physics-informed modeling for accurate blind docking.

Despite the progress made in deep learning-based molecular docking, there is an ongoing debate about whether deep learning could surpass traditional docking methods. Based on a series of experiments, it has been found that while deep learning models excel in pocket searching, their accuracy in ligand conformation generation lags behind traditional molecular docking methods given the same pockets (Yu et al., 2023). Thus, a natural enhancement strategy to further improve model performances involves adopting a hybrid approach, such as incorporating traditional molecular docking into deep learning models. Some prior efforts (Jeon & Kim, 2020; Ma et al., 2021) have been directed towards such strategy, but exhibit certain limitations, such as operating on two-dimensional (2D) molecular representation rather than directly manipulating three-dimensional (3D) structure.

Bringing traditional paradigms in computational biochemistry into the scope, molecular docking is frequently used in conjunction with molecular dynamics (MD) simulation (Morris & Corte, 2021). This combination allows for a delicate and holistic characterization of protein-ligand interactions. Specifically, molecular docking is used as an initial step to predict several possible binding poses and rank them based on binding affinities. The top-ranked protein-ligand complexes by docking can then be further refined using molecular dynamics simulation to obtain more accurate ligand conformations. However, the computational cost of molecular dynamics simulation is prohibitively high, which makes it infeasible to be used on large datasets.

Inspired by the paradigm in computational biochemistry and recent prominence of diffusion models in generating reasonable ligand conformations for blind docking, we propose an intuitive and effective hybrid approach called DiffSim, which seamlessly integrates MD simulation and diffusion models. We devise a loss function to align ligand trajectories derived from MD simulation and reverse diffusion trajectories sampled from the diffusion model. DiffSim allows the MD-informed diffusion model to harness the strengths of both sides: the physics-based insights as well as conformation refinement capability provided by MD, and the sampling efficiency of the diffusion model.

Intuitively, our hybrid approach DiffSim can help mitigate potential challenges in drug discovery which limits the capacity of diffusion models to generate accurate ligand conformations, encompassing two major aspects that are applicable to a broader range of deep generative models as well:

- **DiffSim leverages MD simulation to introduce conformation flexibility into diffusion models.** Most existing models treat proteins as static and regard ligands as rigid bodies with limited degrees of freedom, which could fail to capture the inherent dynamics of proteins and intricate conformational change of ligands. By integrating MD simulation into diffusion models, DiffSim incorporates the dynamic nature of both proteins and ligands with flexibility at atomic resolution, moving beyond the limitations of static and rigid structure.

- **DiffSim explicitly infuses knowledge of protein-ligand interaction provided by MD simulation into diffusion models.** Previous diffusion models as well as other deep-learning based blind docking models make simplifications of protein-ligand interactions, typically considering only atomic distances and atom types, which render them inadequate for incorporating complex

protein-ligand interactions. Hybridization of diffusion models and MD simulation empowers Diff-Sim to embed intricate protein-ligand interaction inherent in MD simulation.

DiffSim combines the strengths of MD simulations and diffusion models in order to achieve accurate and efficient blind docking. In addition to achieving complementarity between the two, we theoretically showcase the consistency of their dynamics, which further substantiates the rationale for DiffSim to seamlessly integrate MD simulation and diffusion models by aligning their trajectories. Specifically, we demonstrate that diffusion models are, in their essence, coarse-grained simulators for MD simulation, which could be a strong backup for our hybrid approach DiffSim as well as a physics-based perspective for the recent success of diffusion-based molecular generative models.

Empirically, DiffSim demonstrates superior performance over existing models in terms of commonly used evaluation metrics.

To summarize, the main contributions of our work are:

- We propose a novel hybrid approach for accurate and efficient blind docking named DiffSim, which seamlessly integrates diffusion models and MD simulation by aligning reverse diffusion trajectories and MD simulation trajectories. To strike a balance between MD computational cost and the need for conformation refinement, we introduce an active learning approach to selectively curate an MD trajectory dataset based on the training set.
- To the best of our knowledge, this work is the first to explicitly unveil the theoretical connection between diffusion-based molecular conformation generation and MD simulation, which serves as a physics-based support for the alignment strategy adopted by DiffSim as well as a new perspective on recent success of diffusion-based molecular generative models.
- We devise a loss function to guide the reverse diffusion trajectories sampled from the diffusion model to align with the trajectories of MD simulation. Experimental results show that DiffSim have state-of-the-art capability of generating accurate ligand conformations for blind docking.

## 2 RELATED WORK

### 2.1 DEEP GENERATIVE MODELS FOR LIGAND CONFORMATION GENERATION

Ligand conformation generation is essentially a conditional generation task, which requires the ligand to bind to a specific protein. Graph- (Tan et al., 2022) and SMILES-based (Skalic et al., 2019; Xu et al., 2021) generative models produce topological (2D) structures of molecules conditioned on protein context. Although 3D molecular conformations can be obtained by combining 2D generative models with conformation generation modules (Grow et al., 2019; Nguyen et al., 2020), which embed 2D structures into 3D space, choosing suitable embedding algorithms can be quite tricky.

Numerous recent works have turned to 3D conditional generative models to directly produce reasonable binding poses. While these models offer exciting possibilities, they also present significant challenges in modelling the complex protein-ligand interactions. Ragoza et al. (2022) generate pocket-aware 3D molecules by voxelizing molecules in atomic density grids, but ligands and protein pockets are separately encoded. DiffAb (Luo et al., 2022) adopts a pairwise embedding MLP to encode the Euclidean distances and dihedral angles between amino acids, but ignore atomic-level intricacy. TargetDiff (Guan et al., 2023), EQUIBIND (Stärk et al., 2022) and TANKBind (Lu et al., 2022) encode the interaction between the ligand atoms and the protein atoms with GNNs, but only take atomic distances and atom types into account. Besides, most existing models make simplifications by confining ligands and proteins to quite limited degrees of freedom. For example, DiffSBDD (Schneuing et al., 2022) and DiffDock (Corso et al., 2022) keeps the protein fixed throughout the generation procedure, which does not align well with the actual binding process where both ligands and proteins exhibit conformational flexibility.

Our approach DiffSim leverages MD simulation to go beyond rigid docking and explicitly captures the intricate protein-ligand interaction at atomic resolution.

### 2.2 HYBRID APPROACHES FOR BLIND DOCKING

Despite the advances in deep learning models for blind docking, questions have emerged regarding whether they are superior to traditional approaches. Yu et al. (2023) find that deep learning models

including Stärk et al. (2022); Lu et al. (2022); Corso et al. (2022) excel in searching protein pockets while traditional methods like Yu et al. (2022) are better at docking on given pockets.

Integration of deep learning models and traditional methods opens up a promising avenue to tackling the challenges of blind docking. For example, Zhang et al. (2021); Gomes et al. (2022); Yang et al. (2022) use molecular docking ad molecular dynamics simulations to further screen the ligands generated by deep learning models. However, these approaches simply use traditional methods after the calculations of deep generative models. Thus, the final selected ligands are not always optimal.

Some research directly incorporates traditional methods into the generative process of deep generative models. SBMolGen (Ma et al., 2021) generates ligand molecules by a SMILES-based search tree and RNN, evaluates them by molecular docking, and the docking results are fed back into the search tree to update the weightings. MORLD (Jeon & Kim, 2020) employs docking tools to calculate binding affinities, which are then integrated as rewards in a reinforcement learning framework to generate and optimize ligands. Nevertheless, both SBMolGen and MORLD operate on the SMILES representation of ligands and utilize RDKit (Landrum, 2006) and OpenBabel (O'Boyle et al., 2011) to generate 3D conformation of the ligands, rather than directly manipulating 3D structure.

Our method DiffSim is a hybrid approach seamlessly combining physics-based molecular dynamics simulation with a diffusion model, which directly generates 3D structures. DiffSim involves molecular dynamics simulation to interfere the generative process of the diffusion model.

## 3 METHOD

This section is organized as follows: In Section 3.1, we provide an overview of relevant concepts: MD simulation, score-based generative model (referred to as diffusion model in this paper), and DiffDock (a state-of-the-art diffusion model for blind docking we build our experiments upon). Section 3.2 outlines our hybrid approach DiffSim and the proposed loss function for alignment. In Section 3.3, we introduce an active learning approach to select a subset of training data for MD simulation. Section 3.4 delves into the theoretical connection of MD simulation and diffusion models. We demonstrate that diffusion models can serve as coarse-grained simulators for MD simulation and provide a bias-variance tradeoff perspective on DiffSim.

### 3.1 PRELIMINARIES

#### 3.1.1 MOLECULAR DYNAMICS SIMULATION

We consider an MD trajectory $\mathcal{M}$ of a molecule with T time frames. $\mathcal{M}^{(t)} = \left( \boldsymbol{x}^{(t)}, \boldsymbol{v}^{(t)} \right)$ denotes a snapshot at time $t \in [T]$. There $\boldsymbol{x}^{(t)} \in \mathbb{R}^{N \times 3}$ and $\boldsymbol{v}^{(t)} \in \mathbb{R}^{N \times 3}$ denote the 3D coordinates and the velocities of each atom, respectively.

MD simulations simulate physical movements of atoms in a system and generate the atomic trajectories with certain initial conditions and boundary conditions. This is obtained by solving the first-order differential equation of the Newton's second law: $\boldsymbol{F}_i^{(t)} = m_i \boldsymbol{a}_i^{(t)} = -\frac{\partial U\left(\boldsymbol{x}^{(t)}\right)}{\partial \boldsymbol{x}_i^{(t)}}$ where $\boldsymbol{F}_i^{(t)}$ is the net force acting on the $i$-th atom of the system at a given point in the $t$-th timeframe, $m_i$ is the mass, $\boldsymbol{a}_i^{(t)}$ is the acceleration and $U\left(\boldsymbol{x}\right)$ is the potential energy function defined by force field. Numerical methods are utilized to advance the trajectory over small time increments $\Delta t$ with the assistance of some integrator.

#### 3.1.2 SCORE-BASED GENERATIVE MODEL

Score-based generative models (Song & Ermon, 2019), referred to as diffusion models in this paper, have garnered increasing interest in molecular generation (Ganea et al., 2021).

**Diffusion process.** Assume a diffusion process $\{\boldsymbol{x}(s)\}_{s=0}^{S}$ indexed by a continuos time variable $s \in [0, S]$, such that $\boldsymbol{x}(0) \sim p_0$, for which we have a dataset of i.i.d. samples, and $\boldsymbol{x}(S) \sim p_S$, for which we have a tractable form to generate samples efficiently. Let $p_s(\boldsymbol{x})$ be the probability density of $\boldsymbol{x}(s)$. The diffusion process is modeled as the solution to an Itô SDE:

$$\mathrm{d}\boldsymbol{x} = f(\boldsymbol{x}, s)\mathrm{d}s + g(s)\mathrm{d}\boldsymbol{w}, \tag{1}$$

where $\boldsymbol{w}$ is a standard Wiener process, $f(\cdot, s) : \mathbb{R}^d \to \mathbb{R}^d$ is a vector-valued function called the drift coefficient of $\boldsymbol{x}(s)$, and $g(\cdot) : \mathbb{R} \to \mathbb{R}$ is a scalar function known as the diffusion coefficient of $\boldsymbol{x}(s)$.

**Reverse process.** By starting from samples of $\boldsymbol{x}(S) \sim p_S$ and reversing the diffusion process, we can obtain samples $\boldsymbol{x}(0) \sim p_0$. The reverse-time SDE is also a diffusion process as:

$$\mathrm{d}\boldsymbol{x} = \left[ f(\boldsymbol{x}, s) - g(s)^2 \nabla_{\boldsymbol{x}} \log p_s(\boldsymbol{x}) \right] \mathrm{d}s + g(s) \mathrm{d}\overline{\boldsymbol{w}}, \tag{2}$$

where $\overline{\boldsymbol{w}}$ is a standard Wiener process when time flows backwards from $S$ to $0$, and $ds$ is an infinitesimal negative timeframe. The score of a distribution can be estimated by training a score-based model on samples with score matching.

### 3.1.3 DIFFDOCK

DiffDock (Corso et al., 2022) is a state-of-the-art diffusion model for blind docking, upon which we conduct our experiment. A diffusion process is defined over translations, rotations, and torsion angles of the ligand. Translations of the ligand position, rigid rotations of the ligand, changes in torsion angles at each rotatable bond are associated with the 3D translation group $\mathbb{T}(3)$, the 3D rotation group $SO(3)$, and a copy of 2D rotation group $SO(2)$ repectively. In all three groups, the forward SDE is defined as $dx = \sqrt{d\sigma^2(t)/dt}d\boldsymbol{w}$ where $\sigma^2 = \sigma_{\mathrm{tr}}^2, \sigma_{\mathrm{rot}}^2$, or $\sigma_{\mathrm{tor}}^2$ for $\mathbb{T}(3)$, $SO(3)$, $SO(2)$ respectively and where $\boldsymbol{w}$ is the corresponding Brownian motion.

### 3.2 DIFFSIM STRATEGY: SEARCH AND REFINE IN A UNIFIED FRAMEWORK

With a seed ligand conformation generated by RDKit ETKDG (Landrum et al., 2013) as input, we train DiffSim by aligning the reverse diffusion trajectory of the conformation with its MD simulation trajectory to generate accurate ligand conformations for blind docking. Specifically, we consider there are $N$ iteration steps, and denote the conformation sampled from the reverse diffusion process at step $i$ as $C_i^{\mathrm{rev}}$ and the conformation optimized through MD simulation at step $i$ as $C_i^{\mathrm{sim}}$. The loss function for alignment is as:

$$L = \sum_{i=1}^{N} w_i \mathrm{RMSD}(C_i^{\mathrm{rev}}, C_i^{\mathrm{sim}}), \tag{3}$$

where $\mathrm{RMSD}(C_i^{\mathrm{rev}}, C_i^{\mathrm{sim}}) = \sqrt{\frac{1}{n} \sum_{j=1}^{n} ||\mathbf{r}_j^{\mathrm{rev}} - \mathbf{r}_j^{\mathrm{sim}}||^2}$, with the total number of atoms denoted as $n$, the position vectors of the $j$-th atom of the reverse diffusion process and MD simulation process denoted as $\mathbf{r}_j^{\mathrm{rev}}$ and $\mathbf{r}_j^{\mathrm{sim}}$, respectively. To encourage the desired alignment similarity, we devise different weighting schedules in experiments, including equal weighting $w_i = \frac{1}{N}$ and gradual weighting $w_i = \frac{e^{i/N}}{\sum_{j=1}^{N} e^{j/N}}$. Equal weighting assigns each iteration step with equal importance while gradual weighting emphasizes alignments at later stages since the conformation gradually converges to the ground truth over time, with later steps providing more accurate representations.

However, a critical prerequisite for alignment is that the diffusion model and MD simulation exhibit comparable sampling efficiencies. This poses a significant challenge in aligning the two due to the inherent discrepancy between their sampling efficiencies. Directly aligning them from the initial conformation would result in a noticeable lag in the sampling efficiency of MD simulation compared to the diffusion model. Empirically, the diffusion model can reach the same bounded conformation within a mere 20 steps, whereas millions of steps of all-atom MD simulations would be required to achieve the same outcome (Lu et al., 2023).

To tackle such challenge and facilitate meaningful alignment between the diffusion model and MD simulation, DiffSim utilizes the diffusion model alone in the initial few steps of reverse diffusion before conducting MD-informed reverse diffusion through the loss function for alignment. The initial few steps serve the purpose of "pocket searching", rapidly exploring the vast conformational space and narrowing it down to a limited subspace. The subsequent MD-informed steps are intended for "conformation refinement" to achieve more accurate and physics-informed ligand conformations. With the subspace as the starting point for alignment, MD simulation can better keep pace with the diffusion model and fully exert the conformation refinement capabilities.

### 3.3 ACTIVE LEARNING-BASED SELECTION FOR MD SIMULATION

Despite a powerful technique to investigate the structural and dynamical protein-ligand properties, MD simulation consumes considerable computational resources (Liu et al., 2016). For example, one-microsecond simulation of a relatively small system running on 24 processors requires months of computation to complete (Durrant & McCammon, 2011), and expensive petascale supercomputers are required to study larger systems (Sothiselvam et al., 2014). The substantial computational cost of MD simulation makes it practically infeasible to apply on the entire training set.

To strike a balance between computational resources and the need for meticulous conformation refinement, we introduce an active learning approach to select a subset from the training data, for which MD simulation can be most effectively leveraged. In particular, we choose ligands whose final conformations, after reverse diffusion, exhibit root mean square deviation (RMSD) values exceeding 2Å, while maintaining centroid distance values below 1Å. 1Å, 2Å, and 5Å have been the most commonly used thresholds in the literature(Lu et al., 2022; Stärk et al., 2022; Corso et al., 2022; Yu et al., 2023), and from the practical aspect of computational cost, we consider the number of ligands that satisfy these criteria, leading to the choice of 1Å and 2Å as thresholds for centroid distance and RMSD, respectively. This criterion signifies the successful identification of approximate binding site locations, yet highlights the need for further conformation refinement.

### 3.4 DIFFSIM AS INTERMEDIARY BETWEEN NEWTONIAN AND LANGEVIN DYNAMICS

We leverage the connection between Newtonian dynamics and Langevin dynamics to offer theoretical insights into DiffSim's alignment of MD simulation and diffusion model. We demonstrate diffusion models and Langevin dynamics are mathematically equivalent so that diffusion models can be conceptualized as carrying out coarse-grained MD simulations. Further, we formulate a generalized bias-variance decomposition and develop a bias-variance tradeoff perspective on DiffSim.

**Diffusion Models: Coarse-grained Simulators for MD Simulation**
In MD simulation, the motions of all atoms are described by Newton's second law (i.e., Newtonian dynamics). When it comes to a small subset of degrees of freedom, the Langevin equation serves as a successful description of the dynamics in such subspace (i.e., Langevin dynamics). Langevin dynamics can be regarded as Newtonian dynamics with effective force fields and added frictional forces as well as (time-dependent) fluctuating forces. In particular, assume we would like to consider motion along a small subset of the whole coordinate space defined by the coordinates $q_1, ..., q_M$ for $M \ll N$. The Langevin equations which model the dynamics in this subspace are then:

$$\mu_j \frac{d^2}{dt^2} q_j = -\frac{\partial}{\partial q_j} W(q_1, \ldots, q_M) - \gamma_j \frac{d}{dt} q_j + \sigma_j \xi_j(t) (j = 1, 2, ..., M), \qquad (4)$$

The first term on its r.h.s. describes the force field of an effective potential $W(q_1, \ldots, q_M)$, the second term describes the velocity dependent frictional forces, and the third term the fluctuating forces $\xi_j(t)$ with coupling constants $\sigma_j$. $W(q_1, \ldots, q_M)$ includes the effect of the thermal motion of the remaining $n - M$ degrees of freedom on the motion along the coordinates $q_1, \ldots, q_M$.

**Lemma 3.1.** *Diffusion process, the reverse process, and Langevin equation, as defined in Equation 1, 2, and 4, respectively, are mathematically equivalent in form.*

*Proof.* Consider stochastic differential equations in the form of a first order differential equation

$$\partial_t \boldsymbol{x}(t) = \boldsymbol{A}[\boldsymbol{x}(t), t] + \mathbf{B}[\boldsymbol{x}(t), t] \cdot \boldsymbol{\eta}(t), \qquad (5)$$

subject to the initial condition $\boldsymbol{x}(0) = \boldsymbol{x}_0$.

Note that diffusion process 1 and reverse process 2 are special cases of 5.

The Langevin equation 4, written here in the form $\mu \ddot{q} = f(q) - \gamma \dot{q} + \sigma \xi(t)$, is also a special case of 5. In fact, defining $\boldsymbol{x} \in \mathbb{R}^2$ with components $x_1 = m\dot{q}, x_2 = mq$ reproduces 5 if one defines

$$\boldsymbol{A}[\boldsymbol{x}(t), t] = \begin{pmatrix} f(x_2/m) - \gamma x_1/m \\ x_1 \end{pmatrix}, \quad \mathbf{B}[\boldsymbol{x}(t), t] = \begin{pmatrix} \sigma & 0 \\ 0 & 0 \end{pmatrix}, \text{ and } \quad \boldsymbol{\eta}(t) = \begin{pmatrix} \xi(t) \\ 0 \end{pmatrix}.$$

We thus conclude that diffusion process, reverse process, and Langevin equation are mathematically equivalent in form as Equation 5. ☐

We showed that the diffusion model is driven by Langevin dynamics, which captures the dynamics within a subspace encompassing only a fraction of all degrees of freedom. The stochastic terms serve to describe the effect of the remaining degrees on the degrees of interest. Specifically, in our experiments which employ DiffDock as the diffusion model, the subset of degrees refers to translational, rotational, and torsional degrees of freedom. Since Langevin dynamics is a modification of Newtonian dynamics for handling constrained degrees of freedom, one can conceptualize diffusion models as carrying out MD simulations within a subset of degrees of freedom.

**Balancing Bias and Variance: A Tradeoff Perspective on DiffSim**
We further show the rationale for aligning MD simulation, representing Newtonian dynamics, and diffusion models, characterized by Langevin dynamics, from a bias-variance tradeoff perspective.

Let $C^{\text{gt}} \in \mathbb{R}^{N \times 3}$ denotes the ground-truth ligand conformation, and $\{C_i\}_{i=1}^N \sim \mathcal{N}(\mu, \Sigma)$ denotes $N$ initial conformation drawn randomly from an isotropic Gaussian distribution, which undergo the reverse diffusion and MD simulation, resulting in $N$ final conformations $\{C_i^{\text{rev}}\}_{i=1}^N$ and $\{C_i^{\text{sim}}\}_{i=1}^N$, respectively. $\phi$, $\widehat{\varphi_i}^{\text{rev}}$, and $\widehat{\varphi_i}^{\text{sim}} \in \mathbb{R}^{3N}$ are defined as $vec(C^{\text{gt}})$, $vec(C^{\text{rev}})$ and $vec(C^{\text{sim}})$, respectively, where $vec$ refers to the vectorization of a multi-dimensional tensor by concatenating its last dimension into a single vector. The bias and variance of $\widehat{\varphi_i}$ are as $\text{Bias}(\widehat{\varphi_i}) := \mathbb{E}\{\widehat{\varphi_i} - \phi\}$, $\text{Cov}(\widehat{\varphi_i}) := \mathbb{E}\{(\widehat{\varphi_i} - \mathbb{E}\{\widehat{\varphi_i}\})(\widehat{\varphi_i} - \mathbb{E}\{\widehat{\varphi_i}\})^T\}$, where $\widehat{\varphi_i}$ denote a generic form of $\widehat{\varphi_i}^{\text{rev}}$ and $\widehat{\varphi_i}^{\text{sim}}$. Define $\overline{\varphi} := [\widehat{\varphi_1} \quad \widehat{\varphi_2} \quad ... \quad \widehat{\varphi_N}]$. We also overload the definition of bias by defining $\text{Bias}(\overline{\varphi}) := [\mathbb{E}\{\widehat{\varphi_1} - \phi\} \quad \mathbb{E}\{\widehat{\varphi_2} - \phi\} \quad ... \quad \mathbb{E}\{\widehat{\varphi_N} - \phi\}]$

We formulate a generalized bias-variance decomposition in Lemma 3.2 (proof in Appendix A).

**Lemma 3.2.** *For a positive semi-definite weighting matrix $W$ and a matrix $\mathbf{X}$, we define $\|\mathbf{X}\|_W^2 := \mathbf{X}^T W \mathbf{X}$. The following generalized bias-variance decomposition holds as*

$$n \sum_i \mathbb{E}\left\{RMSD_i^2\right\} = \text{Tr}\{W \sum_i \text{Cov}(\widehat{\varphi_i})\} + \text{Tr}\{\|\text{Bias}(\overline{\varphi})\|_W^2\}$$

*where $n$ denotes the total number of atoms within the ligand.*

**Remark.** *DiffSim uses active learning to select ligands with small centroid distances but large RMSD values. This selection criteria is analogous to choosing generated conformations having small mean square error with repect to their ground-truth counterparts, when measured by a diagonal weighting matrix $W$ that assigns greater weights to heavy atoms and smaller weights to light atoms. Interestingly, such interpretation of $W$ can be extended to MD simulation and the diffusion model. In both contexts, lighter atoms tend to exhibit greater mobility compared to heavier ones, which makes the conformation refinement process lean more towards the lighter atoms, resulting in ligand conformations that closely align with their ground-truth counterparts when the measurement is defined with $W$ placing greater weights on light atoms. Let $\{\widehat{\varphi_i}^0\}_{i=1}^N$ denote inital conformations, $\{\widehat{\varphi_i}^d\}_{i=1}^N$ as generated conformations after reverse diffusion, $\{\widehat{\varphi_i}^a\}_{i=1}^N$ as selected conformations after active learning, and $\{\widehat{\varphi_i}^m\}_{i=1}^N$ as final conformations after MD-informed reverse diffusion. We have $\sum RMSD(\widehat{\varphi_i}^m) \leq \sum RMSD(\widehat{\varphi_i}^a) \leq \sum RMSD(\widehat{\varphi_i}^d) \leq \sum RMSD(\widehat{\varphi_i}^0)$.*

We hypothesize that $\widehat{\varphi}^{\text{sim}}$ exhibits lower bias but higher variance while $\widehat{\varphi}^{\text{rev}}$ displays higher bias but lower variance, ultimately presenting a dynamic tradeoff between bias and variance within DiffSim. To elaborate, by offering a delicate characterization of protein-ligand interaction at atomic resolution, MD simulation is expected to guide the system towards an ideal conformation with lower bias. However, due to its deterministic nature, distinct initial conformations can lead to divergent conformations, hence higher variance in final outcomes. Conversely, the diffusion model introduces additional stochastic terms of Langevin dynamics and explores a conformation space with a subset of degrees of freedom. While the stochastic terms and the reduced set of degrees may introduce a relatively high bias by diminishing the likelihood of locating the optimal conformation, they lead to more stable outcomes, resulting in lower variance.

In computational biochemistry, attempts have been made to use Langevin dynamics for MD simulation in order to reduce the computational cost and improve numerical stability of MD simulation (Lange & Grubmüller, 2006; Leimkuhler & Matthews, 2013; Paquet et al., 2015). The numerical stability issue, to some extent, supports our hypothesis that MD simulation can exhibit high variance, while Langevin dynamics (diffusion model) help mitigate the variance.

Furthermore, active learning can also play a role in bias-variance tradeoff. It selects the samples with small centroid distances but large RMSDs, which indicates high bias but probably low variance (since the approximate binding site has been identified). MD simulation can then focus on these samples and fit them with low-bias regularization to further reduce the mean square error.

# 4 EXPERIMENT

## 4.1 DATASET

We follow the approach used in prior research (Corso et al., 2022; Stärk et al., 2022) by evaluating DiffSim on PDBBind dataset (Liu et al., 2017) v2020, a collection of protein-ligand structures sourced from Protein Data Bank (PDB) (Berman et al., 2003), and employing time-based splits to create training, validation, and test sets. Specifically, 17,347 complexes discovered before 2019 are for training and validation. 125 unique proteins are randomly selected from 1512 complexes discovered in 2019 or later, and 363 new complexes containing these proteins are as the test set.

We select 711 complexes from the aforementioned training set to curate the MD trajectory dataset through the active learning approach described in section 3.3.

## 4.2 MD SIMULATION SETUP

MD simulations are performed on each pair of the selected ligand conformations and their corresponding target proteins. We use CHARMM36 forcefield (Lee et al., 2016) and GPU-accelerated GROMACS (Van Der Spoel et al., 2005), a MD software for simulations of proteins, lipids, and nucleic acids. Ligands and proteins are preprocessed to prepare their MD simulation input files. Specifically, Ligand hydrogenation is carried out by ProteinsPlus (Schöning-Stierand et al., 2022) server[1]. Ligand topology and parameter files are generated using ACPYPE (Sousa da Silva & Vranken, 2012) Server[2]. Missing protein residues are fixed by Swiss-PdbViewer (Guex & Peitsch, 1997).

The preprocessed protein-ligand complex is solvated in a truncated periodic TIP3P water box, and the minimum distance from the box surface to the complex atoms is set to 12 Å. Counter ions are added to neutralize systems, followed by energy minimization to reduce unreasonable collisions of atoms within the complex. Subsequently, a 100 ps NVT (constant volume and temperature) simulation is conducted at 310K (human body temperature), followed by a 100 ps NPT (constant pressure and temperature) simulation at standard atmospheric pressure before initiating MD simulation.

Following Min et al. (2022), we exclude complexes with certain conditions which cause failed MD simulations: 1) complicated ligands with failed forcefield parametrization; 2) proteins with too many missing residues to repair; and 3) specific types of proteins unsuitable for simulation in a water box.

Consequently, MD simulation is conducted on 412 protein-ligand complexes. For each complex, 10 ns simulation was performed, and 20 snapshots were sampled to form a trajectory.

## 4.3 EVALUATION

We compare our approach DiffSim with existing deep learning-based blind docking models on commonly used evaluation metrics, namely ligand RMSD and centroid distance. Ligand RMSD calculates root-mean-square deviation of corresponding atomic positions between a generated ligand and its ground-truth counterpart. Centroid distance is defined as the Euclidean distance between the averaged coordinate of a generated ligand and its ground-truth counterpart.

Both DiffSim and DiffDock sample 40 initial conformations from an isotropic Gaussian distribution and obtain 40 ligand conformations through reverse diffusion. Among the 40 conformations, the one with the lowest RMSD is designated as "top-1" and the five conformations with the lowest RMSDs as "top-5". In contrast, TANKBind and EQUIBIND generate a single result, equivalent to directly generating the top-1 ligand. As shown in the Tab. 1 and Tab. 2, DiffSim demonstrates superior performance on both RMSD and centroid distance over existing models, which indicates that DiffSim is capable of generating more accurate ligand conformations.

---

[1] https://proteins.plus
[2] https://www.bio2byte.be/acpype/

Table 1: Top-1 PDBBind Docking. For EQUIBIND (Stärk et al., 2022) and TANKBind (Lu et al., 2022), we report the best-performing variants on each metric as described in their paper. DiffSim (equal weight) and DiffSim (gradual weighting) refer to DiffSim trained using the alignment loss function with an equal weighting schedule and a gradual weighting schedule, respectively. DiffSim-w/o search refers to directly aligning reverse diffusion with MD trajectory from the initial conformation. DiffSim-w/o AL refers to random selection for MD simulation without active learning.

| | Ligand RMSD | | | | | Centroid Distance | | | | |
| | Percentiles ↓ | | | % below thresh. ↑ | | Percentiles ↓ | | | % below thresh. ↑ | |
| Methods | 25th | 50th | 75th | 5Å | 2Å | 25th | 50th | 75th | 5Å | 2Å |
|---|---|---|---|---|---|---|---|---|---|---|
| EQUIBIND (/|+Q|+Q2|+S) | 2.1 | 5.6 | 10.3 | 46.4 | 24.6 | 0.9 | 2.0 | 6.2 | 71.0 | 50.6 |
| TANKBind (/|-R|-C|-P) | 2.4 | 4.0 | 7.7 | 61.7 | 19.6 | 1.0 | 1.7 | 4.2 | 77.4 | 56.5 |
| DiffDock | 1.35 | 2.52 | 4.58 | 77.35 | 41.16 | 0.79 | 1.86 | 5.00 | 74.86 | 53.87 |
| DiffSim-w/o search | 2.30 | 3.28 | 5.72 | 62.30 | 36.85 | 0.82 | 1.94 | 5.43 | 75.92 | 50.30 |
| DiffSim-w/o AL | 1.39 | 2.52 | 4.76 | 75.43 | 39.40 | 0.68 | 1.97 | 3.10 | 81.79 | 65.24 |
| DiffSim (equal weight) | **1.30** | 2.47 | **4.50** | **77.90** | 40.33 | 0.45 | **0.97** | 2.29 | **85.91** | 71.82 |
| DiffSim (gradual weight) | 1.31 | **2.44** | 4.65 | 77.62 | **41.71** | **0.44** | **0.97** | **2.14** | 85.64 | **72.10** |

Table 2: Top-5 PDBBind Docking

| | Ligand RMSD | | | | | Centroid Distance | | | | |
| | Percentiles ↓ | | | % below thresh. ↑ | | Percentiles ↓ | | | % below thresh. ↑ | |
| Methods | 25th | 50th | 75th | 5Å | 2Å | 25th | 50th | 75th | 5Å | 2Å |
|---|---|---|---|---|---|---|---|---|---|---|
| DiffDock | **1.52** | 2.88 | 5.43 | 72.10 | 37.40 | 0.50 | 1.13 | 2.50 | 83.37 | 68.34 |
| DiffSim-w/o search | 1.67 | 3.29 | 5.62 | 70.34 | 35.28 | 0.56 | 1.39 | 2.94 | 80.10 | 62.42 |
| DiffSim-w/o AL | 1.62 | 3.00 | 5.37 | 71.97 | 35.19 | 0.53 | 1.22 | 2.56 | 83.32 | 66.50 |
| DiffSim (equal weight) | **1.52** | 2.93 | **5.20** | **73.70** | 37.07 | **0.48** | **1.10** | 2.50 | **84.09** | 67.96 |
| DiffSim (gradual weight) | **1.52** | **2.85** | 5.34 | 72.60 | **37.51** | 0.50 | 1.12 | **2.43** | 83.70 | **68.45** |

Compared to DiffDock, DiffSim exhibits a more pronounced advantage in top-1 ligand than top-5 ligands, which reflects the bias-variance tradeoff perspective discussed in Section 3.4. This experimental observation can be attributed to the incorporation of MD simulation, which increases the likelihood of achieving ideal conformations (low bias) while introduces higher variance in final outcomes. Consequently, DiffSim exhibits a more substantial advantage for top-1 ligand than top-5 ligands whose performances can be affected by variance.

## 4.4 ABLATION STUDY

We conduct ablation studies to investigate factors that influence the performance of DiffSim. As shown in Tab. 1 and Tab. 2, directly aligning reverse diffusion with MD trajectory from the initial conformation (denoted as 'DiffSim-w/o search') causes noticeable decreases in performances, which emphasizes the necessity of 'pocket searching' in the initial few steps by exlusively using the diffusion model and highlights DiffSim's strategy of integrating search and refine in a unified framework. Replacing the active learning approach by random selection (denoted as 'DiffSim-w/o AL') also results in suboptimal outcomes, which indicates that prioritizing informative samples is crucial in harnessing the full potential of MD simulation and facilitating meaningful integration of diffusion model and MD simulation.

## 5 CONCLUSIONS

We propose a new hybrid approach called DiffSim, which aligns reverse diffusion and MD simulation trajectory. DiffSim facilitates diffusion-based generation of accurate ligand conformations, augmented by physics-informed MD insights that transcend the constraints of static and rigid structures and incorporate intricate dynamics of protein-ligand interactions. We demonstrate the theoretical connection between diffusion models and MD simulation, which provides theoretical underpinnings for DiffSim's alignment strategy and inspires a new perspective for recent success of diffusion-based molecular generative models. Empirically, DiffSim demonstrates state-of-the-art capability of generating accurate ligand conformations for blind docking. DiffSim is anticipated to bring fresh insights to other model architectures beyond diffusion models in AI-enhanced drug discovery.

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

# A    PROOF

## A.1    PROOF OF LEMMA 3.2

**Lemma 3.2.** *For a positive semi-definite weighting matrix $W$ and a matrix $\mathbf{X}$, we define $\|\mathbf{X}\|_W^2 := \mathbf{X}^T W \mathbf{X}$. The following generalized bias-variance decomposition holds as*

$$n \sum_i \mathbb{E}\left\{RMSD_i^2\right\} = \mathrm{Tr}\{W \sum_i \mathrm{Cov}(\widehat{\varphi}_i)\} + \mathrm{Tr}\{\|\mathrm{Bias}(\overline{\varphi})\|_W^2\}$$

*where $n$ denotes the total number of atoms within the ligand.*

*Proof.*

$$\begin{aligned}
n\mathbb{E}\left\{\mathrm{RMSD}_i^2\right\} =& \mathbb{E}\left\{\|\widehat{\varphi}_i - \phi\|_W^2\right\} \text{ (recall that $\phi$ refers to the ground-truth conformation)} \\
=& \mathbb{E}\left\{(\widehat{\varphi}_i - \phi)^T W(\hat{\varphi}_i - \phi)\right\} \\
=& \mathbb{E}\left\{\mathrm{Tr}\left\{W(\widehat{\varphi}_i - \phi)(\widehat{\varphi}_i - \phi)^T\right\}\right\} \\
=& \mathrm{Tr}\left\{W\mathbb{E}\left\{(\widehat{\varphi}_i - \phi)(\widehat{\varphi}_i - \phi)^T\right\}\right\} \\
=& \mathrm{Tr}\left\{W\mathbb{E}\left\{(\widehat{\varphi}_i - \mathbb{E}\{\widehat{\varphi}_i\} + \mathrm{Bias}(\widehat{\varphi}_i))(\widehat{\varphi}_i - \mathbb{E}\{\widehat{\varphi}_i\} + \mathrm{Bias}(\widehat{\varphi}_i))^T\right\}\right\} \\
=& \mathrm{Tr}\left\{W\,\mathrm{Cov}(\widehat{\varphi}_i) + W(\mathrm{Bias}(\widehat{\varphi}_i))(\mathrm{Bias}(\widehat{\varphi}_i))^T\right. \\
& + W\mathbb{E}\{(\widehat{\varphi}_i - \mathbb{E}\{\widehat{\varphi}_i\})\}(\mathrm{Bias}(\widehat{\varphi}_i))^T \\
& \left. + W(\mathrm{Bias}(\widehat{\varphi}_i))\mathbb{E}\left\{(\widehat{\varphi}_i - \mathbb{E}\{\widehat{\varphi}_i\})^T\right\}\right\} \\
=& \mathrm{Tr}\left\{W\,\mathrm{Cov}(\widehat{\varphi}_i) + W(\mathrm{Bias}(\widehat{\varphi}_i))(\mathrm{Bias}(\widehat{\varphi}_i))^T\right\} \\
=& \mathrm{Tr}\{W\,\mathrm{Cov}(\widehat{\varphi}_i)\} + \|\mathrm{Bias}(\widehat{\varphi}_i)\|_W^2.
\end{aligned}$$

The fourth equality exploits the linearity of the trace to commute it with the expectation value. The fifth equality uses the identity $\phi = \mathbb{E}\{\hat{\varphi}\} - \mathrm{Bias}(\hat{\varphi})$. The seventh equality uses the fact that $\mathbb{E}\{\hat{\varphi} - \mathbb{E}\{\hat{\varphi}\}\} = 0$.

Sum L.H.S and R.H.S of the above derivation over $i$, and note that $\sum_i \|\mathrm{Bias}(\widehat{\varphi}_i)\|_W^2 = \mathrm{Tr}\{\|\mathrm{Bias}(\overline{\varphi})\|_W^2\}$, thus completing the proof. □

