# OpenReview forum: "DiffSim: Aligning Diffusion Model and Molecular Dynamics Simulation for Accurate Blind Docking"
_ICLR.cc/2024/Conference — Submitted to ICLR 2024_

### Official Review · Reviewer_HbHG · 2023-10-31

**Soundness:** 3 good
**Presentation:** 3 good
**Contribution:** 2 fair
**Rating:** 3
**Confidence:** 5

**Summary:**

This paper proposes DiffSim, a denoising diffusion model of MD simulation for protein-ligand docking. The authors create an MD dataset themselves, which is another critical contribution to the community.

**Strengths:**

- This paper studies an important question of MD simulation.
- The authors generate the MD dataset, which is quite valuable.

**Weaknesses:**

I think one main concern of this paper is the lack of related works and baselines. Other comments may come next after this is solved.

I can understand the technical novelty of DiffSim if we follow the DiffDock research line. However, from the AI for MD literature, there have been several published works on using denoising diffusion for MD simulation [1,2,3]. They are not cited and compared in this work, and I would like to know the authors’ feedback on this.

[1] Wu, Fang, and Stan Z. Li. "DIFFMD: a geometric diffusion model for molecular dynamics simulations." Proceedings of the AAAI Conference on Artificial Intelligence. Vol. 37. No. 4. 2023.

[2] Arts, Marloes, et al. "Two for one: Diffusion models and force fields for coarse-grained molecular dynamics." Journal of Chemical Theory and Computation 19.18 (2023): 6151-6159.

[3] Fu, Xiang, et al. "Simulate Time-integrated Coarse-grained Molecular Dynamics with Multi-scale Graph Networks." Transactions on Machine Learning Research (2023).

**Questions:**

See above.

---

> ### Author Response · Authors · 2023-11-21
>
> Dear Reviewer HbHG,
>
> Thank you for your feedback and for bringing the AI for MD literature to our attention.
>
> Our primary motivation for developing this method was to focus on the integration of symbolic software and neural networks to enhance docking accuracy and introduce a sense of trustworthiness. We recognize the importance of AI for MD, particularly in terms of its time efficiency and other advancements. And we will refine our work by incorporating both branches of research.
>
> Thank you!

---

### Official Review · Reviewer_gbXf · 2023-10-31

**Soundness:** 3 good
**Presentation:** 3 good
**Contribution:** 3 good
**Rating:** 5
**Confidence:** 3

**Summary:**

The paper proposes an interesting hybrid approach for blind docking by integrating diffusion models and molecular dynamics (MD) simulations. The motivation of combining these two methods to achieve accurate and efficient blind docking is reasonable.

**Strengths:**

1. The idea of aligning reverse diffusion and MD trajectories is novel and has potential.
2. The paper demonstrates that the diffusion process, the reverse process, and Langevin equation are mathematically equivalent in form.
3. The paper proposes a loss function to align reverse diffusion sampling with MD simulation trajectories.

**Weaknesses:**

1. The results look a little better than diffdock. The result only improves significantly on the Centroid Distance metric for TOP-1.
2. Due to the choice of DiffDock as the backbone model, its limitations in the degrees of freedom seem conflict with the philosophy of MD.
3. One benchmark is not enough to show the effectiveness of the method.

**Questions:**

1. What if MD combined with an all-atom diffusion model such as Geodiff [1], it feels that an all-atom diffusion method is more compatible with MD.
2. Do the authors try different RMSD thresholds for active learning?
3. Has the author tried comparing the efficiency of DiffSim with traditional docking tools and other deep learning methods?
4. I believe DiffSim is designed to enable flexible binding docking. Perhaps it would be beneficial to incorporate visualization features that display protein changes, such as side chain alterations, during the process.
5. Also, what is the pocket RMSD? or if you just use holo protein/pocket as initial docking?
5. It might be valuable to include a discussion on the time efficiency of molecular dynamics (MD) simulations in this stage, particularly in the context of protein MD.

[1] Xu, Minkai, et al. "GeoDiff: A Geometric Diffusion Model for Molecular Conformation Generation." International Conference on Learning Representations. 2021.

---

> ### Author Response · Authors · 2023-11-21
>
> Dear Reviewer gbXf,
>
> Thank you for your comments.
>
> We use holo protein as initial conformation for protein, thus not considering protein RMSD.
>
> Combining MD with an all-atom diffusion model such as Geodiff is an interesting idea. While our paper focuses on aligning reverse diffusion and MD trajectories, exploring the compatibility and potential benefits of integrating all-atom diffusion models with MD is an avenue worth investigating.
>
> Admittedly, MD simulation does take considerable time. Time efficiency is the major focus we would like to improve in future work.
>
> Thank you for your time and efforts in reviewing our paper!

---

### Official Review · Reviewer_jMvQ · 2023-11-03

**Soundness:** 2 fair
**Presentation:** 3 good
**Contribution:** 3 good
**Rating:** 3
**Confidence:** 4

**Summary:**

The paper proposes DiffSim as a hybrid framework of protein docking. DiffSim first uses a diffusion model alone to do a quick "pocket search". Then DiffSim aligns the reverse diffusion sampling trajectories with the simulation trajectories from MD. It uses active learning to select a subset of training samples for MD simulation.

**Strengths:**

The general topic of utilizing MD to improve neural docking method is interesting.

The entire framework is significant in contribution, and the improved performance is satisfying for stats on RMSD and Top 1 acc.

The introduction of "bias and variance" between MD and reverse diffusion is interesting.

**Weaknesses:**

1. The discussion between the connection of diffusion an MD is weak, and the proposed aligning method seems not to be solid. Lemma 3.1 simply states that the reverse diffusion process has the same differential form of a Langevin dynamics. This is known since 2020. But, the per-step marginal distributions of reverse dynamics should be very much different, therefore a step-wise bounding between reverse diffusion and MD traj is not solid in theory. More specifically, Langevin dynamics sample from an Boltzmann distribution under the energy function, while intermediate reverse diffusion steps sample from a Gaussian, changing as $t$ evolves. This introduce a variable "energy" (log probability) in reverse diffusion models. Authors would refer to [1] [2] or [3] for deeper discussions between diffusion and MD.

2. Lack of visualization, which is vital for MD analysis. Authors should provide consistent dynamics of reverse diffusion to justify that the trajectories are correctly learnt.

3. The hypothesis of "bias and variance" tradeoff is not supported by any results. Authors should report the recorded metrics as they can be easily calculated.

[1] Two for One: Diffusion Models and Force Fields for Coarse-Grained Molecular Dynamics. https://arxiv.org/abs/2302.00600

[2] Score-Based Generative Modeling through Stochastic Differential Equations. https://arxiv.org/abs/2011.13456

[3] Towards Predicting Equilibrium Distributions for Molecular Systems with Deep Learning. https://www.microsoft.com/en-us/research/publication/towards-predicting-equilibrium-distributions-for-molecular-systems-with-deep-learning/

**Questions:**

1. I'd like to see more visualization results and studies of empirical evidence of "bias and variance" tradeoff.

2. Analysis must be done on the MD trajectories to show that the proposed protocol is reasonable.

3. Authors claim that Top1 acc is improved more significantly than Top5 and credit this to "bias and variance" tradeoff. This needs more justification.

---

> ### Author Response · Authors · 2023-11-18
>
> Dear Reviewer jMvQ,
>
> Thank you for your comments on our work. We appreciate your insights and the references you provided for further discussions in terms of the connection between MD and diffusion.
>
> The step-wise bounding approach may not be optimal. In light of your feedback, we are delving deeper into the literature and consider theoretically compatible approaches to aligning reverse diffusion and MD trajectories, taking into account the differences in sampling distributions.
>
> Once again, we express our gratitude for your time and effort in reviewing our paper.

---

### Official Review · Reviewer_uuBj · 2023-11-03

**Soundness:** 3 good
**Presentation:** 3 good
**Contribution:** 2 fair
**Rating:** 3
**Confidence:** 5

**Summary:**

This paper proposes a new method called DiffSim for accurate blind protein-ligand docking. The key ideas are:

- DiffSim integrates a diffusion-based generative model with molecular dynamics (MD) simulation to combine their strengths.
- It aligns the reverse diffusion sampling process with MD trajectories using a novel loss function.
- An active learning approach selectively chooses training samples for MD simulation.
- Theoretical analysis shows consistency in dynamics between diffusion models and MD, making DiffSim a reasonable hybrid.
- DiffSim outperforms previous blind docking methods on standard RMSD and centroid distance metrics.

**Strengths:**

- Novel idea to seamlessly combine generative diffusion model with MD simulation.
- Theoretical analysis gives useful insights into connections between the two approaches.
- Active learning makes selective use of expensive MD simulation.
- Strong empirical results validate accuracy improvements over state-of-the-art methods.

**Weaknesses:**

- More analysis of alignment loss function forms could be useful, and also ablation study isolating active learning benefits would be informative.
- Testing on more diverse protein-ligand complexes beyond PDBBind. Currently, the benchmark comparison is limited. Some other papers are necessary to be compared. Such as, E3Bind, https://openreview.net/forum?id=sO1QiAftQFv, FABind, https://arxiv.org/abs/2310.06763.
- Computational efficiency comparison to alternatives would be helpful. It is necessary to give computational comparison since MD is usually cost, while diffusion with many steps are also cost.
- The experimental results are not good as expected, which leads to a negative view of the effectiveness of the method.

**Questions:**

NA

---

> ### Author Response · Authors · 2023-11-18
>
> Dear Reviewer uuBj,
>
> Thank you sincerely for your valuable feedback on our paper.
>
> We plan to conduct analysis on the alignment loss function forms in our future research and expand our benchmark comparisons beyond the current scope. Regarding computational efficiency, MD simulations are indeed computationally expensive, and we are exploring faster alternatives that can still provide reliable results.
>
> We are committed to addressing these concerns in our future endeavors.
>
> Thank you once again for your time and efforts in reviewing our work.

---

### Meta-Review · Area_Chair_zM7x · 2023-12-05

**Metareview:**

DiffSim is a novel method proposed for protein-ligand docking, combining a diffusion-based generative model with molecular dynamics (MD) simulation. Its key innovations include the integration of diffusion and MD simulation techniques, a specialized loss function for aligning reverse diffusion sampling with MD trajectories, and an active learning approach for selecting efficient training samples for MD simulation. The method starts with a diffusion model for quick "pocket search" in protein-ligand docking, followed by aligning these search results with MD simulation trajectories. The paper also contributes to the scientific community by creating a new MD dataset.

**Justification For Why Not Higher Score:**

The reviewers raised the following main concerns:
* The paper provides limited insight into the connection between diffusion processes and molecular dynamics, lacking depth in its theoretical underpinnings.
* The evaluation of the method is considered insufficient, not convincingly demonstrating the utility of the approach.
* The discussion on related work, particularly concerning the use of denoising diffusion for MD simulation, is found to be weak, indicating a need for a more comprehensive review and analysis in this area.

**Justification For Why Not Lower Score:**

N/A

---

### Decision · Program_Chairs · 2024-01-16

Reject